# Clinical Significance of Lipoprotein Lipase (LPL) in People Living with HIV: A Comprehensive Assessment Including Lipidemia, Body Composition, Insulin Secretion, and Insulin Resistance

**DOI:** 10.3390/nu17203207

**Published:** 2025-10-13

**Authors:** Akira Matsumoto, Kunio Yanagisawa, Yoshiyuki Ogawa, Takumi Nagasawa, Mayumi Nishiyama, Koji Sakamaki, Akihiro Yoshida, Masami Murakami, Katsuhiko Tsunekawa, Hiroshi Handa, Takao Kimura

**Affiliations:** 1Department of Hematology, Gunma University Graduate School of Medicine, Showa-Machi 3-39-22, Maebashi 371-8511, Gunma, Japan; 2Division of Internal Medicine, Fukaya Red Cross Hospital, Kamishiba-cho-nishi 5-8-1, Fukaya 366-0052, Saitama, Japan; k-yanagisawa.ne@fukaya.jrc.or.jp; 3Blood Transfusion Service, Gunma University Graduate School of Medicine, Showa-Machi 3-39-22, Maebashi 371-8511, Gunma, Japan; 4Department of Clinical Laboratory Medicine, Gunma University Graduate School of Medicine, Showa-Machi 3-39-22, Maebashi 371-8511, Gunma, Japanayossie10@gunma-u.ac.jp (A.Y.);; 5Clinical Laboratory Center, Gunma University Hospital, Showa-Machi 3-39-22, Maebashi 371-8511, Gunma, Japan; 6Centre for Health Control, Hidaka Hospital, Nakao-Machi-886, Takasaki 370-0001, Gunma, Japan; 7Medical Education and Development, Gunma University Graduate School of Medicine, Showa-Machi 3-39-22, Maebashi 371-8511, Gunma, Japan; 8Graduate School of Food and Population Health Science, Gunma University, Aramaki-Machi 4-2, Maebashi 371-8510, Gunma, Japan; 9General Health Support Center, Gunma University, Aramaki-Machi 4-2, Maebashi 371-8510, Gunma, Japan

**Keywords:** cardiovascular diseases, muscle cells, insulin resistance, risk assessment

## Abstract

**Background/Objectives**: Dyslipidemia is one of the major problems of long-term management in people living with human immunodeficiency virus (HIV) (PLH) as a risk factor for cardiovascular diseases. Lipoprotein lipase (LPL) is anchored on the surface of the capillary endothelial cells and plays a pivotal role in triglyceride metabolism by catabolizing dietary chylomicrons and very low-density lipoprotein synthesized in the liver. However, the details of the mechanisms in the era of integrase strand transfer inhibitor-based antiretroviral therapy have not yet been clarified. **Methods**: This study was a cross-sectional, single-center, non-interventional study evaluating the underlying factors associated with dyslipidemia, insulin resistance or secretion, and changes in the body composition of PLH. **Results**: Among PLH (*n* = 48), lower LPL (<60.8 ng/mL) and older age independently predicted antilipemic drug (ALD) necessity. A comparison of ALD-naïve PLH (*n* = 33) and age- and sex-matched non-HIV controls (*n* = 33) showed that PLH were significantly associated with lower high-density lipoprotein cholesterol (HDL-C) and higher HOMA-β. LPL was also the independent predictor of HDL-C < 40 mg/dL in PLH (adjusted odds ratio = 0.901, *p* = 0.044). Furthermore, LPL < 65.3 ng/mL predicted HDL-C < 40 mg/dL with 100% sensitivity and 60.9% specificity. Low levels of HIV-RNA were detected in the high HOMA-β group. **Conclusions**: In Japanese individuals, compared to non-HIV controls, PLH has low HDL-C and LPL. The measurement of LPL may confer the risk assessment and decision-making with relevance to ALD in PLH. Additionally, the effectiveness of HIV antiviral therapy and glucose tolerance may interact with each other.

## 1. Introduction

Human immunodeficiency virus (HIV) infection is a cause of acquired immunodeficiency syndrome. However, the development of antiretroviral treatment (ART) conferred improved survival, expected to reach to equal those of individuals without HIV [1]. Today, the management of long-term complications is similar to that of individuals without HIV; hence, patients with HIV are called “people living with HIV” (PLH). Cardiovascular diseases are essential long-term comorbidities of PLH [2], and arteriosclerosis risk factors, including dyslipidemia, diabetes, hypertension, chronic kidney disease, and weight gain, should be focused as the major problems of PLH [3,4]. In particular, dyslipidemia in PLH is associated with a higher incidence of atherosclerosis-related diseases, such as ischemic heart disease and cerebrovascular disease, compared to healthy controls [5,6], and the prevalence varies from 20% to 80% [7,8,9] depending on the study populations [10]. The causes of dyslipidemia in PLH are thought to be diverse, including ART comprising antiretroviral drugs (ARVs), HIV infection itself, hepatitis virus co-infection [11,12], and lifestyle factors, such as diet and obesity, but the detailed mechanisms are still largely unknown.

In recent years, the analysis of the lipid abnormalities contributing to the development of atherosclerosis has focused on the rise in triglyceride (TG) contained in postprandial chylomicrons and very low-density lipoprotein (VLDL-C) cholesterol and the involvement of proteins, such as lipoprotein lipase (LPL) and hepatic triglyceride lipase (HTGL), which are important for triglyceride metabolism. Glycosylphosphatidylinositol-anchored high-density lipoprotein-binding protein 1 (GPIHBP1) is an anchor protein of LPL in the vascular endothelium, and autoantibodies against GPIHBP1 can decrease the LPL activity levels to cause hypertriglyceridemia [13]. We have reported that insulin resistance leads to a decrease in LPL and an increase in GPIHBP1 levels [14]. The level of LPL in peripheral bloodstream tends to be lower in individuals with more risk factors for metabolic syndrome, such as impaired glucose tolerance, dyslipidemia, and hypertension. In the case of PLH, it is unclear whether low LPL levels are also observed, as seen in metabolic syndrome. As reported, LPL and HTGL may be partially responsible for the lipid abnormalities in PLH at a time when the primary component drugs of ART are protease inhibitors [15]. However, in the recently advanced setting of ART, the significance of regulating molecules, including LPL, has not yet been well examined.

In this study, we aimed to evaluate the underlying factors in the change in lipid abnormality, body composition, insulin resistance, and secretion to clarify the useful indicators for the long-term management of PLH. In addition, we analyzed the relationship between the effectiveness of HIV antiviral therapy and glucose and lipid metabolism.

## 2. Materials and Methods

### 2.1. Participants

The recruitment period of PLH who visited Gunma University Hospital (Gunma, Japan) (*n* = 57) was from 4 August 2023 to 27 December 2024. The control data were collected from the database of individuals who visited the Centre for Health Control of Hidaka Hospital from 17 March 2017 to 31 January 2018 (*n* = 143). This study was conducted using an opt-out method, with a public notice posted on the hospital’s website. Each participant was given the opportunity to decline participation.

Figure 1 shows the study populations. It should be noted that all female participants (*n* = 75) were excluded due to differences in body composition, impact of menopausal status on lipid metabolism, and the fact that most of the PLH participants were male. The participants were instructed to visit after a 12 h fast. One PLH participants who visited after meals were excluded (*n* = 1). Three participants who made duplicate visits (*n* = 3) were excluded as well.

### 2.2. Study Design

This study was a cross-sectional, single-center, non-interventional study that evaluated the underlying factors associated with dyslipidemia, changes in body composition, insulin resistance, and secretion of PLH to clarify useful indicators for their long-term management. For this purpose, we conducted three sub-analyses: (1) an age- and sex-matched analysis comparing PLH and controls (*n* = 33 vs. 33) performed using propensity score matching; (2) a single-population analysis of PLH, including clinical and treatment profiles (*n* = 48); and (3) an analysis of PLH without antilipemic drugs (ALD) and controls (*n* = 33 vs. 33) to exclude the effects of lipid-lowering treatment. All biological sample collections and physical examinations were conducted after a 12 h fast. We do not measure hemoglobin A1c (HbA1c) in PLH without diabetes mellitus (DM). The study was conducted in accordance with the Declaration of Helsinki and approved by the ethical board of Gunma University Hospital (HS2023-011, 8 May 2023). Informed consent from study subjects was obtained by opt-out.

### 2.3. Anthropometric Measurement and Laboratory Analyses

The height and the weight of the participants were measured to calculate the body mass index (BMI) computed as the weight divided by height in meters squared (kg/m^2^). The waist circumference (WC) was measured to the nearest 0.5 cm using a tape measure placed horizontally at the level of the umbilicus while the participants quietly exhaled [16]. A bioimpedance instrument (InBody 470; InBody Japan, Tokyo, Japan) was used to measure the body weight (BW), mass and rate of fat, skeletal muscle mass, and moisture in the standing position. The fat mass index (FMI) and the skeletal muscle index (SMI) were calculated as follows: fat mass/height squared (kg/m^2^), and skeletal muscle mass/height squared (kg/m^2^). All measurements were performed by a well-trained technical staff in our laboratory.

The systolic blood pressure and the diastolic blood pressure were also measured, and the blood samples were collected from the antecubital vein while the participant was sitting after an overnight fast. The serum total cholesterol (TC), high and low-density lipoprotein (HDL, LDL) cholesterol (HDL-C, LDL-C) and TG concentrations were measured using enzymatic assays with an automatic analyzer (TBA-c8000; Canon Medical Systems Corporation, Tochigi Japan). The fasting plasma glucose (FPG) concentrations were assessed using the glucose oxidase method, and the HbA1c levels were examined with high-performance liquid chromatography using automatic analyzers (GA 08 II; A&T and HLC-723G9; Tosoh Corporation, Tokyo, Japan). The red blood cell count, hemoglobin and hematocrit levels, white blood cell count, and platelet count were measured using an automatic analyzer (XT-1800i; Sysmex Corporation, Hyogo, Japan). The serum ferritin (FRTN) concentrations were investigated using a latex-enhanced turbidimetric immunoassay with the same analyzer. The serum insulin concentrations were measured using a chemiluminescence immunoassay (AIA-2000 LA; Tosoh, Tokyo, Japan) [17,18]. The serum LPL concentrations were measured using a sandwich enzyme-linked immunosorbent assay (ELISA, Sekisui Medical Co., Ltd., Tokyo, Japan) [18,19]. The GPIHBP1 and HTGL levels were measured using ELISA (IBL Co., Ltd., Fujioka, Gunma, Japan) [20,21]. The collected serum samples were separated by centrifugation (1500× *g*) at 4 °C for 10 min and stored at −80 °C until analysis.

### 2.4. Statistical Methods

The continuous variables were expressed as median values with an interquartile range because almost all variables were not normally distributed. The HOMA-IR and Homeostasis Model Assessment of β-Cell Function (HOMA-β) scores [22] were calculated. The differences between the two groups in each were analyzed using Mann–Whitney *U*-tests, and Kruskal–Wallis tests were used between multiple comparisons of more than two groups. The categorical variables were expressed with frequencies, and differences between groups were analyzed using chi-square tests. To account for potential confounding among variables, we conducted analyses using both multivariable and univariable logistic regression models. Furthermore, we performed a receiver operating characteristic (ROC) analysis using statistically significant univariates, excluding those used to define the outcome. In the ROC analysis, the variable with the highest area under the curve (AUC) was selected as the optimal variable, and sensitivity and specificity were calculated using the cutoff value yielding the highest Youden index among the selected variables. Differences and correlations were considered significant when two-sided *p*-values < 0.05. All statistical analyses and creating illustrations were performed using IBM SPSS Statistics version 30.0.

## 3. Results

### 3.1. Age- and Sex-Matched Analysis

As shown in Table 1, compared to the control group, PLH exhibited lower LPL levels (56.1 vs. 75.1 ng/mL, *p* = 0.019), lower FPG (97 vs. 107 mg/dL, *p* = 0.011), higher HOMA-β (66.4 vs. 49.8, *p* = 0.027), lower HDL-C levels (53 vs. 57 mg/dL, *p* = 0.021), a higher diabetes prevalence (12.1% vs. 0.0%, *p* = 0.039), and a lower hypertension prevalence (33.3% vs. 60.6%, *p* = 0.026). FPG was less than 126 mg/dL in both the PLH and non-HIV control groups. Among these variables, Figure 2 demonstrates the distributions of LPL, HDL-C, and HOMA-β across four groups considering comparisons within both the control and PLH groups based on the BMI (<25 vs. ≥25 kg/m^2^), triglycerides (TG < 150 vs. ≥150 mg/dL), and LDL-C (<140 vs. ≥140 mg/dL). As a result, in both the control and PLH groups, the HDL-C levels were significantly lower in individuals with BMI ≥ 25 and TG ≥ 150. In PLH, HOMA-β was significantly higher in those with BMI ≥ 25 and TG ≥ 150, while LPL levels were significantly higher in those with LDL-C ≥ 140. Additionally, compared with the control group, PLH exhibited lower LPL and HDL-C levels in individuals with TG < 150 and LDL-C < 140. Furthermore, among those with LDL-C <140, HOMA-β was higher in PLH than in the control group.

These findings suggest that PLH possess underlying metabolic changes, even when TG < 150 mg/dL or LDL-C < 140 mg/dL, which are below the clinical thresholds for dyslipidemia. However, this analysis includes PLH receiving ALD, and the results may be influenced by their effects. Therefore, a further analysis considering the clinical necessity of ALD treatment is warranted.

### 3.2. PLH Single-Population Analysis

As shown in Table 2a,b, Appendix A, we conducted a two-group comparison among the 48 PLH participants based on the following criteria: presence or absence of ALD treatment (*n* = 15 vs. 33), statin use (*n* = 11 vs. 37), fibrate use (*n* = 6 vs. 42), HOMA-β ≥ 80 or <80 (*n* = 18 vs. 30), HOMA-IR ≥ 2.5 or <2.5 (*n* = 11 vs. 37), and BMI ≥ 25 or <25 (*n* = 21 vs. 27).

The ALD treatment group (*n* = 15) was significantly older (52.0 vs. 43.0 years old, *p* = 0.011) and exhibited lower LPL levels (50.6 vs. 65.5 ng/mL, *p* = 0.024), higher HTGL levels (51.4 vs. 41.1 ng/mL, *p* = 0.039), and higher FPG (99.0 vs. 94.0 mg/dL, *p* = 0.020). Additionally, diabetes prevalence was higher (20.0% vs. 3.0%, *p* = 0.049), and dolutegravir (DTG) (53.3% vs. 24.2%, *p* = 0.048) and elvitegravir/cobicistat (13.3% vs. 0.0%, *p* = 0.032) use was more frequent in this group.

The statin user group (*n* = 11) was significantly older (52.0 vs. 43.0 years, *p* = 0.025) and had higher FPG levels (99.0 vs. 95.0 mg/dL, *p* = 0.037). In the fibrate user group (*n* = 6), the LPL levels were significantly lower (43.4 vs. 62.5 ng/mL, *p* = 0.038), while the HOMA-IR values (2.3 vs. 1.4, *p* = 0.042), diabetes prevalence (33.3% vs. 4.8%, *p* = 0.018), and TG levels (178.5 vs. 105.0 mg/dL, *p* = 0.016) were significantly higher.

In the HOMA-β ≥ 80 group (*n* = 18), the participants exhibited significantly higher fasting insulin levels (10.9 vs. 4.1 μU/mL, *p* = 0.001), higher HOMA-IR (2.7 vs. 0.9, *p* < 0.001), higher HOMA-β values (121.1 vs. 51.0, *p* < 0.001), higher TG levels (145.0 vs. 98.0 mg/dL, *p* = 0.012), higher BW (79.6 vs. 68.0 kg, *p* = 0.021), higher BMI (26.9 vs. 23.7 kg/m^2^, *p* = 0.002), and higher FMI (7.7 vs. 5.6 kg/m^2^, *p* = 0.003). These findings suggest that in PLH, those with high HOMA-β values exhibit increased basal insulin secretion as a compensatory mechanism to counteract insulin resistance. Increased insulin is needed to maintain proper blood glucose levels. Additionally, they showed a lower SMI/FMI ratio (1.1 vs. 1.4, *p* = 0.007), higher body fat percentage (27.7% vs. 21.7%, *p* = 0.002), larger WC (93.4 vs. 83.2 cm, *p* = 0.003), and a lower viral suppression rate (HIV-RNA < 20 copies/mL: 88.9% vs. 96.7%, *p* = 0.011).

In the HOMA-IR ≥ 2.5 group (*n* = 11), the participants exhibited significantly higher fasting insulin levels (15.1 vs. 5.6 μU/mL, *p* < 0.001), higher FPG (99.0 vs. 94.0 mg/dL, *p* = 0.003), higher HOMA-IR values (4.0 vs. 1.4, *p* < 0.001), higher HOMA-β values (151.0 vs. 56.0, *p* < 0.001), and higher TG levels (163.0 vs. 106.0 mg/dL, *p* = 0.048). Additionally, they had greater BW (84.6 vs. 68.2 kg, *p* = 0.001), higher BMI (27.3 vs. 23.8 kg/m^2^, *p* < 0.001), higher SMI (8.6 vs. 7.8 kg/m^2^, *p* = 0.001), higher FMI (8.2 vs. 5.9 kg/m^2^, *p* = 0.004), lower SMI/FMI ratio (1.1 vs. 1.4, *p* = 0.019), higher body fat percentage (28.1% vs. 22.8%, *p* = 0.003), and a larger WC (96.5 vs. 83.2 cm, *p* < 0.001).

Finally, in the BMI ≥ 25 group (*n* = 21), the participants exhibited significantly higher fasting insulin levels (9.3 vs. 4.2 μU/mL, *p* < 0.001), higher HOMA-IR values (2.3 vs. 0.9, *p* < 0.001), higher HOMA-β values (99.3 vs. 52.4, *p* = 0.002), higher TG levels (140.0 vs. 96.0 mg/dL, *p* = 0.013), and higher LDL-C levels (124.0 vs. 112.0 mg/dL, *p* = 0.020). Additionally, BW, BMI, SMI, FMI, body fat percentage, and WC were all significantly higher (*p* < 0.001), while the SMI/FMI ratio was significantly lower (1.0 vs. 1.6, *p* < 0.001).

To adjust for confounding among variables showing significant differences in the two-group comparisons, we performed both multivariable and univariable logistic regression analyses, with each of the following as the dependent variable: ALD use, statin use, fibrate use, HOMA-β ≥ 80, HOMA-IR ≥ 2.5, and BMI ≥ 25. In these analyses, the variables included in the definition of the dependent variable or were considered to have a direct proportional relationship were excluded from the analysis. The analysis results are presented in Table 3.

In the multivariable analysis, the significant variables for ALD use were age (adjusted odds ratio (aOR) 1.17, 95% confidence interval (CI) 1.04–1.31, *p* = 0.010) and LPL (aOR 0.91, 95% CI 1.85–1.98, *p* = 0.014).

For fibrate use, the significant variables included HOMA-IR (aOR 0.06, 95% CI 0.00–0.87, *p* = 0.039) and a history of diabetes (aOR 4128.10, 95% CI 2.89–5.9 × 10^6^, *p* = 0.025).

Regarding BMI ≥ 25, IRI (aOR 1.18, 95% CI 1.03–1.35, *p* = 0.018) and LDL-C (aOR 1.06, 95% CI 1.01–1.10, *p* = 0.010) were also identified as significant variables.

For HOMA-β ≥ 80, TG (aOR 1.01, 95% CI 1.00–1.03, *p* = 0.030) was found to be significant. No significant variables were identified for HOMA-IR ≥ 2.5.

We conducted an additional ROC analysis using the significant variables identified in the previous univariates analysis results, because of the significance of multivariable analysis after Bonferroni-adjusted *p*-values (Appendix A). Consequently, the optimal cutoff values for ALD use were 49.5 years of age (AUC 0.730, *p* = 0.004, sensitivity 60.0%, specificity 75.8%, Figure 3a) and an LPL level of 60.8 ng/mL (AUC 0.711, *p* = 0.007, sensitivity 85.7%, specificity 62.5%, Figure 3b).

Similarly, for statin use, the optimal age cutoff was 47.5 years (AUC 0.724, *p* = 0.006, sensitivity 72.7%, specificity 64.9%, Figure 3c), while for fibrate use, the optimal TG cutoff was 106.5 mg/dL (AUC 0.800, *p* < 0.001, sensitivity 100.0%, specificity 54.8%, Figure 3d).

For BMI ≥ 25, the optimal cutoff values were TG 114.0 mg/dL (AUC 0.710, *p* = 0.008, sensitivity 66.7%, specificity 74.1%, Figure 3e) and LDL-C-C 97.0 mg/dL (AUC = 0.698, *p* = 0.009, sensitivity 100.0%, specificity 37.0%, Figure 3f).

For HOMA-IR and HOMA-β as dependent variables, the TG levels and body composition indices were generally significant explanatory variables (AUC ≥ 0.7, *p* = 0.001, Figure 3g,h).

For HOMA-IR ≥ 2.5, the BMI was identified as the most optimal explanatory variable, with the highest AUC of 0.880 (*p* < 0.001) and an optimal cutoff value of 26.5 (sensitivity 90.9%, specificity 86.5%). In contrast, for HOMA-β ≥ 80, the BMI was not the most optimal variable; instead, the body fat percentage had the highest AUC of 0.775 with an optimal cutoff of ≥25.3, yielding a sensitivity of 83.3% and a specificity of 73.3%.

### 3.3. Analysis of PLH Without ALD and Controls

As shown in Table 4a,b, we included PLH not receiving ALD (*n* = 33) and compared various indices with HIV-negative controls (*n* = 33). First, although PLH patients were significantly younger (43.0 vs. 51.0 years old, *p* = 0.001) and had a significantly lower hypertension prevalence (18.2% vs. 60.6%, *p* < 0.001), they exhibited significantly lower HDL-C levels (50.0 vs. 57.0 mg/dL, *p* = 0.021), higher HOMA-β values (64.2 vs. 49.8, *p* = 0.038), and lower FPG levels (94.0 vs. 107.0 mg/dL, *p* < 0.001). These facts suggest that PLH patients are young but have high insulin resistance. Therefore, we conducted subgroup comparisons within PLH based on indicators of insulin resistance, HOMA-IR ≥ 2.5 (*n* = 11) vs. <2.5 (*n* = 22), HOMA-β ≥ 80 (*n* = 12) vs. <80 (*n* = 21), and HDL-C <40 mg/dL vs. ≥40 mg/dL (*n* = 6 vs. 23, four PLH were not assessed for HDL-C levels).

In the subgroup analysis among PLH, the HOMA-IR ≥ 2.5 group (*n* = 11) exhibited significantly higher fasting insulin levels (IRI, 16.3 vs. 4.9 μU/mL, *p* = 0.008), higher HOMA-IR values (4.0 vs. 1.1, *p* = 0.005), and higher HOMA-β values (163.0 vs. 57.1, *p* = 0.008). Additionally, in the HOMA-IR ≥ 2.5 group, the BMI was significantly higher (29.1 vs. 22.8, *p* = 0.047), and the prevalence of BMI ≥ 25.0 was also significantly higher in this group (85.7 vs. 30.8%, *p* = 0.009). Moreover, the prevalence of TG ≥ 150 mg/dL was significantly higher (71.4 vs. 11.5%, *p* = 0.001).

In the HOMA-β ≥ 80 group (*n* = 12), the fasting insulin levels were significantly higher (11.8 vs. 3.9 μU/mL, *p* < 0.001), and both HOMA-IR (2.7 vs. 0.9, *p* < 0.001) and HOMA-β (133.0 vs. 50.1, *p* < 0.001) were significantly elevated. Additionally, the TG levels and the prevalence of TG ≥ 150 mg/dL were significantly higher in this group (135.5 vs. 82.0 mg/dL, *p* = 0.008; 50.0 vs. 9.5%, *p* = 0.009, respectively). The BMI was significantly higher (27.0 vs. 22.8 kg/m^2^, *p* = 0.040), along with a significantly higher prevalence of BMI ≥ 25 (66.7 vs. 28.6%, *p* = 0.030), and the body fat percentage was also significantly increased (29.5 vs. 20.7%, *p* = 0.010). Additionally, the WC was significantly larger (95.3 vs. 81.8 cm, *p* = 0.020).

Finally, in the HDL-C < 40 mg/dL group (*n* = 6), the HDL-C (32.0 vs. 56.0 mg/dL, *p* < 0.001) and LPL (46.1 vs. 69.7 ng/mL, *p* = 0.039) levels were significantly lower compared to those in the HDL-C ≥ 40 mg/dL group. However, no significant differences were observed between the two groups in other serum lipid parameters, markers of insulin secretion or resistance, and body composition indices.

Finally, LPL was the only significant variable for HDL-C < 40 mg/dL is important. Excluding the classical serum lipid markers (i.e., TG and LDL-C) used in the decision-making for ALD administration, the results of the multivariate and univariate logistic regression models using four explanatory variables—LPL, HOMA-IR, HOMA-β, and BMI—with HDL-C < 40 mg/dL as the dependent variable are shown in Table 5. Consequently, even after considering the confounding effects of other variables, LPL was a significant explanatory variable for HDL-C < 40 mg/dL in PLH (aOR 0.90, 95% CI 0.81–1.00, *p* = 0.044). Furthermore, the ROC analysis was conducted with LPL as the explanatory variable and HDL-C < 40 mg/dL as the dependent variable. The results showed that the optimal cutoff level for LPL was 65.3 ng/mL (AUC 0.800, *p* = 0.001, sensitivity 100.0%, specificity 60.9%, Figure 4).

## 4. Discussion

In this study, we investigated the metabolic characteristics of PLH, focusing on LPL, body composition, and the relationship of LPL with lipid abnormalities, insulin resistance, and treatment necessity. Even when TG, HDL-C, and LDL-C were within clinically normal ranges, LPL and HDL-C in PLH seemed to be associated with lower than non-HIV controls. Among PLH, LPL < 60.8 ng/mL and older age (≥49.5-year-old) independently predicted ALD necessity. Compared to non-HIV controls, ALD-naïve PLH were associated with lower HDL-C, LPL, and higher HOMA-β, even with lower incidence of hypertension.Compared to non-HIV controls, PLH have insulin resistance reflected by low plasma HDL-C and serum LPL. LPL was the independent predictor of HDL-C < 40 mg/dL in PLH (aOR = 0.901, *p* = 0.044). Furthermore, LPL < 65.3 ng/mL predicted HDL-C < 40 mg/dL with 100% sensitivity and 60.9% specificity, which may predict the cardiovascular risk associated with lower HDL-C. Therefore, even when classical lipid markers were within clinically normal ranges, the LPL measurement may confer risk assessment and decision-making with relevance to ALD in PLH. Additionally, low levels of HIV-RNA were detected in the high HOMA-β group. These indicate that HIV antiviral therapy and glucose metabolism may interact with each other.

In this study, in PLH, data were found that reflected increased insulin resistance, with low HDL-C and low LPL in peripheral blood stream. Compared with non-HIV controls, PLH had lower LPL and HDL-C levels and higher HOMA-β. In PLH, the higher the BMI and TG levels, the lower the LPL and HDL-C, but the higher the HOMA-β. We recently reported that greater insulin resistance was associated with lower levels of LPL and HDL-C, but not higher HOMA-β, in non-HIV controls [14]. We reported that HOMA-β decreases with age in Japanese individuals without HIV [23]. HOMA-β reported in our previous reports is comparable to that in non-HIV controls in this study [14,23]. Although four DM patients were included, high insulin resistance and high HOMA-β may be characteristics of male PLH. In PLH, patients with high HOMA-βlevels were characterized by high insulin resistance, as indicated by high HOMA-IR, fasting insulin, body fat mass, WC, BMI, and TG, and low LPL and HDL-C. These findings suggest that in PLH, those with high HOMA-β values exhibit increased basal insulin secretion as a compensatory mechanism to counteract insulin resistance. Increased insulin is needed to maintain proper blood glucose levels. These facts suggest that the high HOMA-β in PLH in this study is due to increased insulin resistance. Recent studies in USA showed a higher prevalence of diabetes mellitus in PLH [24], and studies in China showed a value of triglyceride-glucose (TyG) index, an emerging marker of insulin resistance to predict the incidence of type 2 diabetes mellitus (T2DM) [25]. These conflicts between them and our results suggest the difference in genetic background due to ethnicity and lifestyle especially in food. PLH with high HOMA-β levels were more likely to have detectable HIV-RNA despite being on HIV treatment. This result is consistent with recent reports from other groups. Recent reports have shown a relationship between poor viral control or low-level viremia and insulin resistance [26] or an increased risk of diabetes mellitus [27]. Furthermore, in human adipose tissue, the integrase inhibitors dolutegravir (DTG) and raltegravir (RAL) induce insulin resistance [28] and suppress LPL mRNA expression [29]. Both DTG and RAL increased the interleukin-6 (IL-6) gene expression [28,29]. Taken together with our findings, the presence of detectable HIV-RNA under ART may be associated with increased accumulation of visceral fat, insulin resistance, and IL-6 and decreased mRNA expression of LPL in adipose tissue.

Most studies that focused on the lipid abnormalities associated with HIV patients have examined the significance of therapeutic agents using classical lipid markers (TG, LDL-C, HDL-C) as indicators; only a few have discussed the mechanisms of dyslipidemia. Recently, a large interventional study reported that the administration of pitavastatin reduced cardiovascular events in PLH [30]. Additional analyses of this study have discussed changes in classical lipid and inflammatory markers, plaques (especially in the coronary artery), and ARVs; however, the contributing factors as causes of dyslipidemia and the way of stratification for the treatment necessity were unclear [31,32]. Furthermore, they have not focused on the contribution of LPL as an underlying factor of lipid metabolism.

Previous studies have reported the association of dyslipidemia in PLH caused by the HIV infection itself, hepatitis viruses, and ARVs. This study highlights the role of LPL as a contributing factor of dyslipidemia in PLH. In the era of the PI-based regimen, ARVs may suppress the LPL expression, thereby resulting in lipid abnormalities [15]. However, since INSTIs have become the primary HIV treatment component, there have been no reports of studies focusing on lowering LPL. Of the PLHs included in this study, 40 of 48 (83.3%) were treated with INSTIs, which is a novelty in elucidating the significance of LPL reduction under these circumstances. Even when classical lipid markers were within clinically normal ranges, the LPL measurement contributes risk assessment and decision-making with relevance to ALD in PLH.

As described above, a combination of lipid abnormalities, insulin resistance and hypersecretion, weight gain, and glucose intolerance are often observed in PLH; however, it is difficult to clarify a single factor to be a cause of them. Furthermore, all PLH are now recommended to be treated as soon as possible [33,34], so that their metabolic impact of ARVs cannot be ruled out. Most of the currently recommended regimens comprise INSTIs, including either bictegravir (BIC), DTG, or RAL [35,36]. In this study, patients taking DTG were slightly more likely to require ALD than BIC and RAL, but in general, no major differences can be found in the pattern of metabolic abnormalities for each regimen. Interestingly, a recent study comparing the effects of these ARVs on adipokines and inflammatory markers from adipocytes in vitro showed significant differences in the profile of molecules induced by each loading drug [29], with DTG slightly reducing the expression of adipogenic marker genes (peroxisome proliferator-activated receptor-Ɣ and lipoprotein lipase) being slightly decreased. Furthermore, both DTG and RAL increased the IL-6 gene expression, but only DTG increased the IL-6 release. In addition, DTG suppressed the adiponectin expression and secretion in differentiated adipocytes and had a similar effect on leptin. Adiponectin stimulates muscle cells to increase LPL release and activity [37,38]. These indicate that DTG and RAL reduce LPL levels through two mechanisms: a direct effect on LPL expression and an indirect effect by inducing insulin resistance through increased levels of the inflammatory cytokine such as IL-6. The results of this study are consistent with these previous reports. In comparison to these results, BIC inhibited gene expression and secretion of proinflammatory cytokines in differentiated adipocytes. As mentioned above, this study suggests that an increase in body fat percentage may affect the effectiveness of ART in reducing viral load. However, evidence for the clinical management of metabolic abnormalities and specific association for each ARVs has not yet been established.

LPL is an enzyme that hydrolyzes TG in the peripheral bloodstream to produce fatty acids and glycerol, which are responsible for supplying energy to the peripheral tissues and storing fat. It is mainly distributed in the adipose tissue, cardiac muscle, and skeletal muscle and involved in the metabolism of TG contained in chylomicrons and VLDL-C [39]. The relationship between HIV and LPL was analyzed in the PI-ART era; however, we could not find a study focused on the association between LPL and the clinical manifestation of PLH in the INSTI-ART era. In clinical practice, we prescribe lipid-lowering drugs, such as statins, fibrates, and EPA preparations, according to the values of classic lipid markers like LDL-C, HDL-C, and TG; however, for example, controlling LDL-C with a statin reduces atherosclerotic disease by only approximately 40%. In other words, the individualization of patient management methods based on precise lipid disorder pathology other than the classic lipid markers is still in the research stage not just for PLH. Serum HDL-C levels were inversely associated with all-cause mortality in the general Japanese population [40]. Among non-HIV individuals, the lower the LPL, the lower the HDL-C [14]. In this study, LDL-C, TG, HDL-C, and LPL levels were at the same levels in the statin-treated and non-treated groups in PLH, indicating that statins were effective. On the other hand, TG levels were significantly higher and LPL levels were significantly lower in the fibrate-treated group compared to the non-fibrate-treated group. These facts indicate that it is difficult to lower TG levels to the reference range by administering fibrates monotherapy in PLH. Our findings suggest the utility of assessing LPL mass in the management of lipid abnormalities in patients with HIV. The results of this study indicate that ARVs in PLH may cause a decrease in LPL levels, either directly or indirectly. Taking together our results with these findings, it may be possible to establish surrogate markers that can predict the manifestation of lipid abnormalities by examining the LPL concentration in a larger number of cases for each type of anti-HIV drug.

This study has several limitations. Firstly, this is a single-center observational study, so that findings from our statistical analysis have a risk of false positive due to the limited sample size which did not fully meet the statistical power of 80% at α = 0.05 and β = 0.20. No multiplicity adjustment was applied in the main analysis. Further research with a larger number of participants in multi-center is needed to elucidate causal relationships. If we can follow the present cohort prospectively, we may be able to examine the relationship between changes in the LPL concentration and activity and specific atherosclerotic and premorbid changes, such as the incidence of cardiovascular events and changes on the carotid echocardiogram. Secondly, the administration of antilipemic and anti-HIV drugs and other medications all depend on the physician’s choice. Thirdly, changes in weight, lipid profile, and metabolic abnormalities are strongly influenced by each patient’s lifestyle, specifically diet intake, amount of exercise, and whether they drink or smoke, which were not controlled for in this study. Fourthly, the PLH group, but not in non-HIV controls, in this study included four diabetic patients. We did not analyze HbA1c levels of study subjects. Fifth, no analysis of data from PLH female was conducted. There are gender differences in LPL levels in peripheral bloodstream and insulin resistance. Future analysis of female PLH is needed.

## 5. Conclusions

In Japanese individuals, compared to non-HIV controls, PLH had lower LPL and HDL-C, and higher HOMA-β. Our findings highlight the potential role of LPL as a key marker in lipid metabolism among PLH, providing novel insights into their long-term management. Low HDL-C and low LPL levels in PLH are likely due to two mechanisms: a direct effect of ARVs on reducing LPL expression, and an indirect effect mediated by increased insulin resistance. Furthermore, the results of this study suggested the association of insulin resistance and the control of HIV-RNA under antiviral therapy. Further studies are needed across the country to examine the efficacy of therapeutic interventions targeting the LPL activity to improve the risks of cardiovascular events in PLH.

## Figures and Tables

**Figure 1 nutrients-17-03207-f001:**
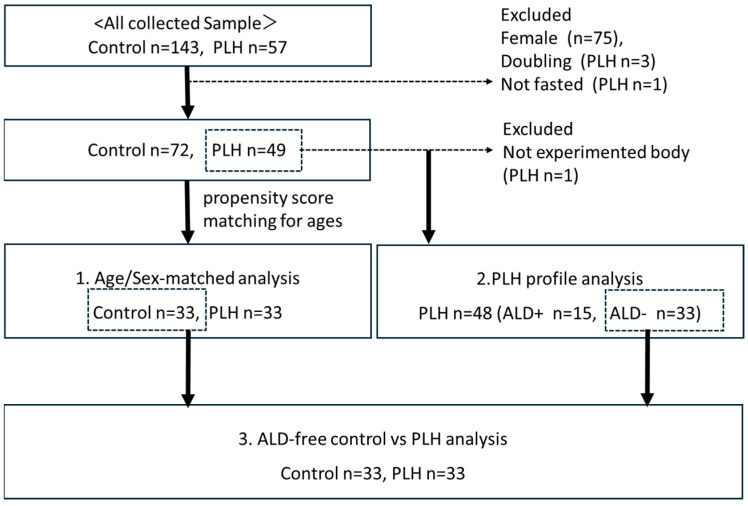
Study population and subgroups.

**Figure 2 nutrients-17-03207-f002:**
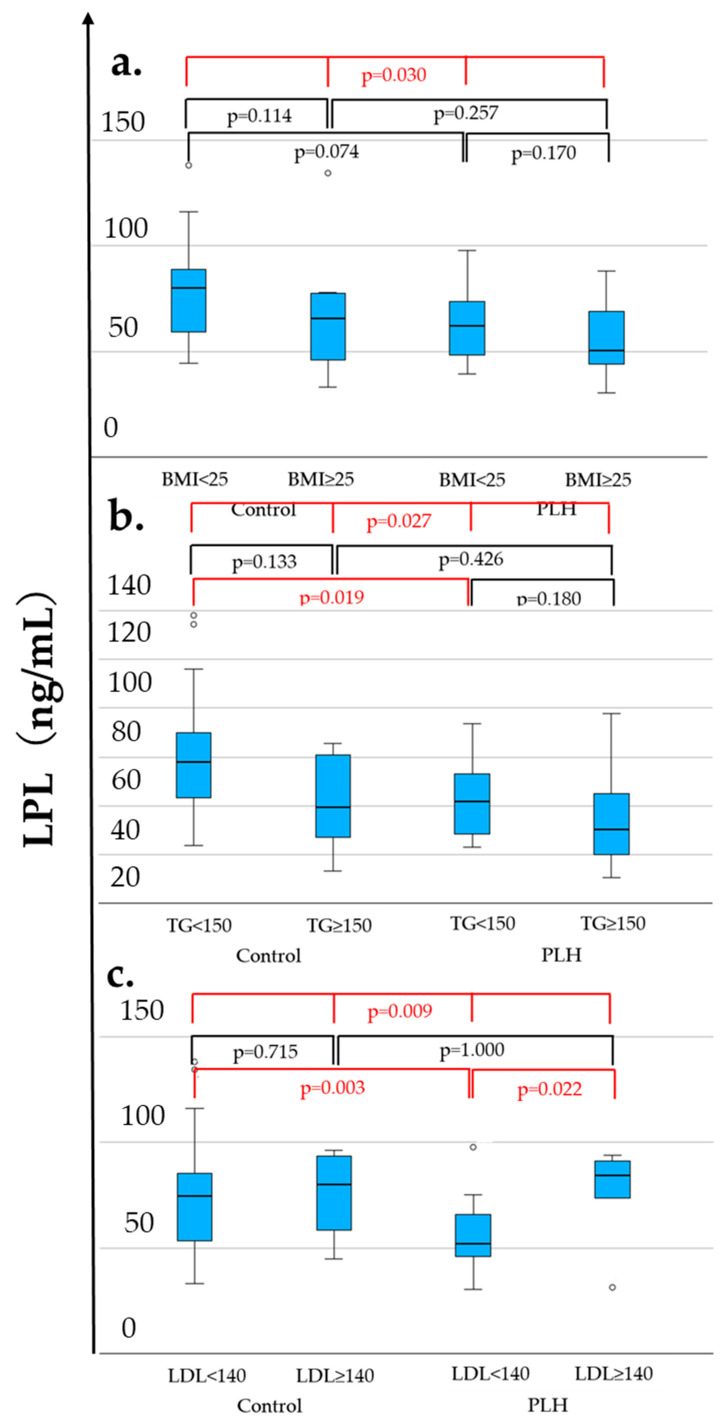
LPL (**a**–**c**), HDL-C (**d**–**f**), and HOMA-β (**g**–**i**) levels associated with the BMI, TG, and LDL-C. The statistically significant differentiation (*p* < 0.05) is colored red.

**Figure 3 nutrients-17-03207-f003:**
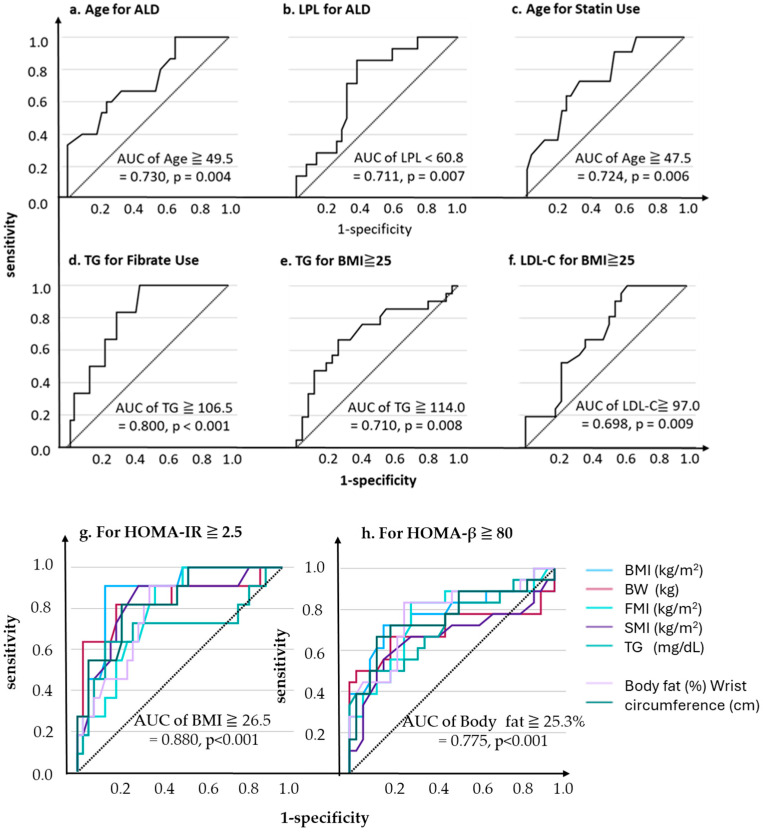
ROC analysis predicting each outcome.

**Figure 4 nutrients-17-03207-f004:**
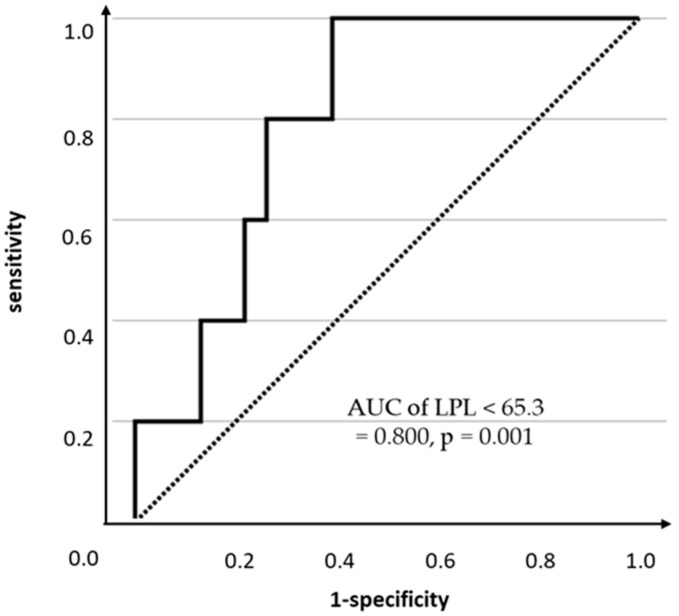
ROC analysis predicting HDL-C < 40 mg/dL by LPL concentration.

**Table 1 nutrients-17-03207-t001:** Characteristics of the age-matched two groups with HIV (PLH) or none (control).

	Control (*n* = 33)		PLH (*n* = 33)		*p*
	Median/*n*	(IQR/%)	Median/*n*	(IQR/%)	
Age	51	(45.0–56.0)	51	(44.0–56.0)	0.933
LPL (ng/mL)	75.1	(53.3–86.9)	56.1	(45.9–72.3)	** 0.019 ** ** * **
GPIHBP1 (pg/mL)	856.1	(683.4–1243.4)	951.0	(740.2–1101.0)	0.537
HTGL (ng/mL)	42.8	(30.8–57.7)	48.4	(37.6–71.8)	0.177
IRI (µU/mL)	5.7	(4.0–7.7)	7.5	(3.8–12.1)	0.339
FRTN (ng/mL)	120.2	(92.9–212.3)	98.6	(66.4–218.8)	0.438
FPG (mg/dL)	107	(98.5–113.0)	97.0	(90.5–107.5)	** 0.011 ** ** * **
HOMA-IR	1.5	(1.0–2.1)	1.7	(0.9–3.0)	0.733
HOMA-β	49.8	(34.2–70.3)	66.4	(48.7–102.6)	** 0.027 ** ** * **
TG (mg/dL)	98	(68.5–166.5)	107	(78.0–162.0)	0.547
HDL-C (mg/dL)	57	(51.0–74.0)	53	(42.5–62.0)	** 0.021 ** ** * **
LDL-C (mg/dL)	120	(101.0–135.5)	117	(102.0–133.5)	0.783
Ht (cm)	169.6	(165.8–173.5)	172.0	(168.0–175.8)	0.466
BW (kg)	69.0	(62.8–77.5)	69.1	(62.9–77.7)	0.859
BMI (kg/m^2^)	23.6	(21.8–25.8)	24.0	(21.2–27.0)	0.906
Body fat (%)	22.6	(19.6–26.7)	25.5	(19.8–29.9)	0.423
WC (cm)	85.0	(81.3–91.3)	84.3	(82.0–92.3)	0.937
Diabetes	0	(0.0)	4	(12.1)	** 0.039 ** ** * **
Hypertension	20	(60.6)	11	(33.3)	** 0.026 ** ** * **
TG ≥ 150 (mg/dL)	10	(30.3)	11	(33.3)	0.792
BMI ≥ 25 (kg/m^2^)	10	(30.3)	14	(43.8)	0.261

Statistically significant differences were indicated with a *p* < 0.05 value and marked with a bold red and *. Abbreviations: IQR—interquartile ranges for median; %—percentages for frequencies; LPL—lipoprotein lipase; GPIHBP1—glycosylphosphatidylinositol-anchored high-density lipoprotein-binding protein 1; HTGL—hepatic triglyceride lipase; IRI—immunoreactive insulin; FRTN—ferritin; FPG—fasted plasma glucose; HOMA-R—Homeostatic Model Assessment for Insulin Resistance; HOMA-β—Homeostatic Model Assessment of β-Cell Function; TG—triglyceride; HDL-C—high-density lipoprotein cholesterol; LDL-C—low-density lipoprotein cholesterol; Ht—height; BW—body weight; BMI—body mass index.

**Table 2 nutrients-17-03207-t002:** (**a**) PLH categorized using an antilipemic drug or not (*n* = 48, summary of positive findings). (**b**) PLH categorized by HOMA-β/-IR, and BMI (*n* = 48, summary of positive findings).

**(a)**																
	**ALD (−) (*n* = 33)**		**ALD (+) (*n* = 15)**		** *p* **	**Statin (−) (*n* = 37)**		**Statin (+) (*n* = 11)**		** *p* **		**Fibrate (−) (*n* = 42)**		**Fibrate (+) (*n* = 6)**		** *p* **
	**Median/*n***	**(IQR/%)**	**Median/*n***	**(IQR/%)**		**Median/*n***	**(IQR/%)**	**Median/*n***	**(IQR/%)**			**Median/*n***	**(IQR/%)**	**Median/*n***	**(IQR/%)**	
Age	43.0	(35.0–50.0)	52.0	(42.0–61.0)	** 0.011 * **	43.0	(35.5–51.5)	52.0	(42.0–61.0)	** 0.025 * **		43.5	(36.8–52.3)	52.0	(39.5–63.3)	0.180
LPL (ng/mL)	65.5	(46.6–79.5)	50.6	(42.4–59.3)	** 0.024 * **	62.8	(45.9–74.6)	50.6	(45.8–61.5)	0.173		62.5	(47.4–74.2)	43.4	(35.4–56.1)	** 0.038 * **
HTGL (ng/mL)	41.1	(30.4–57.4)	51.4	(45.1–63.8)	** 0.039 * **	42.1	(31.1–57.4)	56.0	(45.1–76.7)	0.053		43.6	(31.9–64.0)	50.9	(45.2–60.2)	0.408
FPG (mg/dL)	94.0	(89.0–99.0)	99.0	(95.0–117.0)	** 0.020 * **	95.0	(89.5–99.0)	99.0	(96.0–110.0)	** 0.037 * **		96.0	(90.0–99.3)	101.0	(93.8–117.3)	0.249
HOMA-IR	1.4	(0.9–2.0)	2.1	(1.2–3.1)	0.074	1.6	(0.9–2.2)	2.1	(0.9–3.7)	0.243		1.4	(0.9–2.2)	2.3	(2.0–3.5)	** 0.042 * **
Diabetes	1	(3.0)	3	(20.0)	** 0.049 * **	2	(5.4)	2	(18.2)	0.178		2	(4.8)	2	(33.3)	** 0.018 **
TG (mg/dL)	104.0	(66.0–147.5)	129.0	(100.0–236.0)	0.083	107.0	(71.0–159.5)	107.0	(93.0–134.0)	0.825		105.0	(71.5–144.3)	178.5	(127.3–269.5)	** 0.016 * **
Antilipenmic drug (ALD)	0	(0.0)	15	(100.0)	NA	4	(10.8)	11	(100.0)	** <0.001 * **		9	(21.4)	**6**	(100.0)	** <0.001 * **
Statin	0	(0.0)	11	(73.3)	NA	0	(0.0)	11	(100.0)	NA		9	(21.4)	2	(33.3)	0.516
Fibrate	0	(0.0)	5	(33.3)	NA	4	(10.8)	2	(18.2)	0.516		0	(0.0)	6	(100.0)	NA
DTG	8	(24.2)	8	(53.3)	** 0.048 * **	10	(27.0)	6	(54.5)	0.089		13	(31.0)	3	(50.0)	0.355
EVG/c	0	(0.0)	2	(13.3)	** 0.032 * **	1	(2.7)	1	(9.1)	0.352		1	(2.4)	1	(16.7)	0.101
**(b)**																
	**HOMA-β < 80 (*n* = 30)**		**HOMA-β ≥ 80 (*n* = 18)**		** *p* **	**HOMA-IR < 2.5 (*n* = 37)**		**HOMA-IR ≥ 2.5 (*n* = 11)**		** *p* **		**BMI < 25 (*n* = 27)**		**BMI ≥ 25 (*n* = 21)**		** *p* **
	**Median/*n***	**(IQR/%)**	**Median/*n***	**(IQR/%)**		**Median/*n***	**(IQR/%)**	**Median/*n***	**(IQR/%)**			**Median/*n***	**(IQR/%)**	**Median/*n***	**(IQR/%)**	
IRI (µU/mL)	4.1	(3.0–6.8)	10.9	(8.6–19.4)	** <0.001 * **	5.6	(3.6–7.8)	15.1	(12.9–29.1)	** <0.001 * **		4.2	(2.5–9.5)	9.3	(6.2–14.3)	** <0.001 * **
FPG (mg/dL)	96.0	(90.8–101.0)	96.5	(90.0–101.8)	0.848	94.0	(89.5–99.0)	99.0	(97.0–122.0)	** 0.003 * **		94	(89.0–100.0)	99.0	(92.0–105.5)	0.058
HOMA-IR	0.9	(0.8–1.7)	2.7	(2.0–4.8)	** <0.001 * **	1.4	(0.9–1.9)	4.0	(3.1–11.8)	** <0.001 * **		0.9	(0.6–2.7)	2.3	(1.5–3.9)	** <0.001 * **
HOMA-β	51.0	(38.7–62.7)	121.1	(99.2–164.4)	** <0.001 * **	56.0	(39.3–75.2)	151.0	(99.3–252.0)	** <0.001 * **		52.4	(37.5–103.3)	99.3	(55.0–142.5)	** 0.002 * **
TG (mg/dL)	98.0	(68.0–137.8)	145.0	(101.3–255.3)	** 0.012 * **	106.0	(75.5–138.5)	163.0	(72.0–254.0)	** 0.048 * **		96.0	(70.0–155.0)	140.0	(99.5–236.0)	** 0.013 * **
TG ≥ 150 (mg/dL)	4	(13.3)	**9**	(50.0)	** 0.006 * **	6.0	(16.2)	7.0	(63.6)	** 0.002 * **		3	(11.1)	10	(47.6)	** 0.005 * **
LDL (mg/dL)	120.5	(92.8–136.8)	112.0	(104.8–125.0)	0.632	118.0	(97.0–137.5)	112.0	(104.0–124.0)	0.624		112.0	(90.0–135.5)	124.0	(111.5–137.5)	** 0.020 * **
BW (kg)	68.0	(63.7–73.5)	79.6	(66.4–90.0)	** 0.021 * **	68.2	(63.1–74.3)	84.6	(76.9–95.2)	** <0.001 * **		65.0	(60.3–79.0)	80.0	(73.0–88.1)	** <0.001 * **
BMI (kg/m^2^)	23.7	(21.0–25.3)	26.9	(24.5–29.3)	** 0.002 * **	23.8	(20.9–25.5)	27.3	(26.6–29.3)	** <0.001 * **		21.6	(20.7–27.4)	27.3	(26.3–29.2)	** <0.001 * **
BMI ≥ 25 (kg/m^2^)	8	(26.7)	13	(72.2)	** 0.002 * **	11.0	(29.7)	10.0	(90.9)	** <0.001 * **		0	(0.0)	21	(100.0)	NA
SMI (kg/m^2^)	7.9	(7.3–8.2)	8.4	(7.4–8.8)	0.060	7.8	(7.3–8.2)	8.6	(8.2–9.0)	** 0.001 * **		7.5	(7.1–8.1)	8.5	(8.2–8.9)	** <0.001 * **
FMI (kg/m^2^)	5.6	(3.9–7.4)	7.7	(6.7–10.4)	** 0.003 * **	5.9	(3.9–7.6)	8.2	(6.9–10.9)	** 0.004 * **		4.8	(3.8–8.4)	8.2	(6.8–9.7)	** <0.001 * **
SMI/FMI	1.4	(1.1–1.9)	1.1	(0.9–1.3)	** 0.007 * **	1.4	(1.0–2.0)	1.1	(0.9–1.2)	** 0.019 * **		1.6	(1.1–1.3)	1.0	(0.9–1.3)	** <0.001 * **
Body fat (%)	21.7	(18.0–26.8)	27.7	(25.6–33.4)	** 0.002 * **	22.8	(18.0–28.1)	28.1	(26.4–34.1)	** 0.003 * **		20.7	(17.2–29.1)	28.1	(25.2–32.3)	** <0.001 * **
Waist circumference (cm)	83.2	(79.3–88.3)	93.4	(83.5–100.0)	** 0.003 * **	83.2	(79.2–90.1)	96.5	(91.0–109.1)	** <0.001 * **		81.9	(76.5–93.7)	92.8	(89.0–100.3)	** <0.001 * **
CD4	620.0	(492.3–826.0)	517.0	(457.0–694.8)	0.198	639.0	(481.5–814.5)	508.0	(461.0–673.0)	0.215		599.0	(427.0–730.5)	562.0	(493.5–714.0)	0.843
HIV-RNA < 20 (c/mL)	29	(96.7)	16	(88.9)	** 0.011 * **	35.0	(94.6)	10.0	(90.9)	0.249		25	(92.6)	20	(95.2)	0.121

Statistically significant differences were indicated with a *p* < 0.05 value and marked with a bold red and *. Abbreviations: SMI—skeletal muscle mass index; FMI—fat mass index; CD4—counts of CD4-positive T-lymphocytes; DTG—dolutegravir; EVG/c—elvitegravir combined with cobicistat; NA―Not Applicable. All data include negative findings (no significance between two variables) are listed in the Appendix A.

**Table 3 nutrients-17-03207-t003:** Multivariate and univariate logistic analyses of PLH (*n* = 48).

For ALD (+)	Multivariate Analysis		*p*	Univariate Analysis		*p*
	aOR	(95% CI)		cOR	(95% CI)	
Age	1.17	(1.04–1.31)	** 0.010 * **	1.11	(1.03–1.20)	** 0.008 * **
LPL (ng/mL)	0.91	(1.85–0.98)	** 0.014 * **	0.95	(0.90–0.99)	** 0.019 * **
HTGL (ng/mL)	1	(1.97–1.04)	0.923	1.02	(1.00–1.05)	0.121
FPG (mg/dL)	1.01	(1.95–1.07)	0.829	1.03	(0.99–1.08)	0.103
Diabetes	0.8	(1.02–31.49)	0.903	8	(0.76–84.59)	0.103
DTG or EVG/c	0.51	(1.07–3.61)	0.503	2.48	(0.66–9.34)	0.181
**For Statin (+)**	**aOR**	**(95% CI)**	** *p* **	**crOR**	**(95% CI)**	** *p* **
Age	1.08	(1.99–1.17)	0.069	1.09	(1.01–1.18)	** 0.035 * **
FPG	1.02	(0.98–1.06)	0.37	1.03	(0.99–1.07)	0.144
**For Fibrate (+)**	**aOR**	**(95% CI)**	** *p* **	**crOR**	**(95% CI)**	** *p* **
LPL (ng/mL)	0.95	(0.86–1.06)	0.357	0.92	(0.85–1.01)	0.065
HOMA-IR	0.06	(0.00–0.87)	** 0.039 * **	1.04	(0.83–1.31)	0.741
Diabetes	4128.1	(2.89–5,900,350.40)	** 0.025 * **	10	(1.09–91.44)	** 0.041 * **
TG (mg/dL)	1.04	(1.00–1.09)	0.061	1.01	(1.00–1.02)	** 0.029 * **
**For BMI ≥ 25**	**aOR**	**(95% CI)**	** *p* **	**crOR**	**(95% CI)**	** *p* **
IRI (µU/mL)	1.18	(1.03–1.35)	** 0.018 * **	1.12	(1.00–1.26)	0.061
TG (mg/dL)	1.01	(1.00–1.02)	0.285	1.01	(1.00–1.02)	** 0.022 * **
LDL-C	1.06	(1.01–1.10)	** 0.010 * **	1.04	(1.01–1.07)	** 0.019 * **
**For HOMA-β ≥ 80**	**aOR**	**(95% CI)**	** *p* **	**crOR**	**(95% CI)**	** *p* **
TG (mg/dL)	1.01	(1.00–1.03)	** 0.030 * **	1.01	(1.00–1.03)	** 0.006 * **
BW (kg)	1.12	(0.84–1.50)	0.434	1.08	(1.01–1.15)	** 0.016 * **
BMI (kg/m^2^)	1.11	(0.56–2.22)	0.763	1.38	(1.10–1.74)	** 0.006 * **
FMI (kg/m^2^)	1.82	(0.66–5.02)	0.249	1.54	(1.14–2.10)	** 0.006 * **
Body fat (%)	5.68	(0.22–148.03)	0.297	1.19	(1.06–1.35)	** 0.004 * **
WC (cm)	1.16	(0.85–1.58)	0.344	1.11	(1.03–1.20)	** 0.008 * **
SMI/FMI	0.82	(0.53–1.25)	0.349	0.19	(0.04–0.80)	** 0.024 * **
HIV-RNA < 20 (copies/mL)	0.04	(0.00–1.27)	0.067	0.28	(0.02–3.28)	0.308
**For HOMA-IR ≥ 2.5**	**aOR**	**(95% CI)**	** *p* **	**crOR**	**(95% CI)**	** *p* **
TG (mg/dL)	1.01	(0.99–1.03)	0.196	1.01	(1.00–1.02)	** 0.019 * **
BW (kg)	1.1	(0.68–1.80)	0.691	1.15	(1.05–1.26)	** 0.003 * **
BMI (kg/m^2^)	40.56	(0.67–2459.78)	0.077	1.82	(1.22–2.71)	** 0.003 * **
SMI (kg/m^2^)	0.01	(0.00–82.87)	0.331	6.57	(1.72–25.07)	** 0.006 * **
FMI (kg/m^2^)	0.02	(0.00–2.49)	0.11	1.48	(1.09–2.00)	** 0.012 * **
SMI/FMI	0.2	(0.00–26,078.05)	0.787	0.09	(0.01–0.82)	** 0.033 * **
Body fat (%)	2.09	(0.80–5.42)	0.131	1.22	(1.06–1.41)	** 0.007 * **
WC (cm)	0.72	(0.40–1.30)	0.275	1.15	(1.05–1.27)	** 0.004 * **

Both univariate and multivariate analyses were performed by using logistic regression analysis. Those with a probability of significance (*p* < 0.05) are marked with bold red and *.

**Table 4 nutrients-17-03207-t004:** (**a**) Characteristics of control (*n* = 33) vs. PLH without ALD (*n* = 33). (**b**) Characteristics of control (*n* = 33) vs. PLH without ALD (*n* = 33).

**(a)**										
	**Control (*n* = 33)**		**ALD-Free PLH (*n* = 33)**							
	**Median/*n***	**(IQR/%)**	**PLH All (*n* = 33)**		** *p* **	**HOMA-IR < 2.5 (*n* = 26)**		**HOMA-IR ≥ 2.5 (*n* = 7)**		** *p* **
			**Median/*n***	**(IQR/%)**		**Median/*n***	**(IQR/%)**	**Median/*n***	**(IQR/%)**	
Age	51.0	(45.0–56.0)	43.0	(35.0–50.0)	** 0.001 * **	43.5	(34.8–51.3)	40.0	(35.0–46.0)	0.560
LPL (ng/mL)	75.1	(53.3–86.9)	65.5	(46.6–79.5)	0.168	65.5	(49.8–76.4)	56.5	(42.2–90.1)	0.651
GPIHBP1 (pg/mL)	856.1	(683.4–1243.4)	951.0	(735.3–1092.5)	0.684	935.8	(728.3–1076.7)	1022.4	(754.8–1110.4)	0.651
HTGL (ng/mL)	42.8	(30.8–57.7)	41.1	(30.4–57.4)	0.948	37.8	(29.7–54.0)	53.8	(36.7–68.1)	0.651
IRI (µU/mL)	5.7	(4.0–7.7)	5.8	(3.7–9.1)	0.934	4.9	(3.0–7.4)	16.3	(13.1–41.3)	** 0.008 * **
FRTN (ng/mL)	120.2	(92.9–212.3)	90.5	(59.0–146.0)	0.066	86.3	(58.1–124.5)	100.0	(64.9–383.1)	0.928
FPG (mg/dL)	107.0	(98.5–113.0)	94.0	(89.0–99.0)	** <0.001 * **	92.0	(88.8–97.3)	99.0	(97.0–122.0)	0.073
HOMA-IR	1.5	(1.0–2.1)	1.4	(0.9–2.0)	0.516	1.1	(0.6–1.7)	4.0	(2.9–12.4)	** 0.005 * **
HOMA-β	49.8	(34.2–70.3)	64.2	(39.3–107.7)	** 0.038 * **	57.1	(38.7–72.5)	163.0	(110.1–252.0)	** 0.008 * **
TG (mg/dL)	98.0	(68.5–166.5)	104.0	(66.0–147.5)	0.626	90.0	(60.8–121.3)	163.0	(72.0–254.0)	0.346
HDL-C (mg/dL)	57.0	(51.0–74.0)	50.0	(42.0–62.0)	** 0.021 * **	56.0	(43.0–65.0)	48.0	(33.8–51.5)	0.169
LDL-C (mg/dL)	120.0	(101.0–135.5)	115.0	(97.0–136.0)	0.658	116.0	(94.5–136.8)	112.0	(106.0–124.0)	0.928
Ht (cm)	169.6	(165.8–173.5)	173.0	(168.0–177.0)	0.064	172.0	(168.0–176.3)	180.0	(175.0–181.0)	0.073
BW (kg)	69.0	(62.8–77.5)	70.0	(62.4–81.8)	0.724	67.8	(61.6–74.9)	89.8	(84.6–104.1)	0.073
BMI (kg/m^2^)	23.6	(21.8–25.8)	24.0	(20.9–27.0)	0.852	22.8	(20.6–25.4)	29.1	(26.6–32.1)	** 0.047 * **
Body fat (%)	22.6	(19.6–26.7)	22.8	(18.0–30.1)	0.878	20.9	(17.0–27.3)	32.2	(27.3–37.5)	0.073
WC (cm)	85.0	(81.3–91.3)	82.5	(79.2–94.6)	0.686	81.9	(78.0–88.7)	99.6	(92.5–110.2)	0.073
Diabetes	0	(0.0)	1	(3.0)	0.314	0	(0.0)	1	(100.0)	0.050
Hypertension	20	(60.6)	6	(18.2)	** <0.001 * **	3	(11.5)	3	(42.9)	0.057
TG ≥ 150 (mg/dL)	10	(30.3)	8	(24.2)	0.580	3	(11.5)	5	(71.4)	** 0.001 * **
BMI ≥ 25 (kg/m^2^)	10	(30.3)	14	(42.4)	0.306	8	(30.8)	6	(85.7)	** 0.009 * **
**(b)**										
**ALD-Free PLH (*n* = 33)**										
	**HOMA-β < 80 (*n* = 21)**		**HOMA-β ≥ 80 (*n* = 12)**		** *p* **	**HDL-C ≥ 40 (*n* = 23) ***		**HDL-C < 40 (*n* = 6) ***		** *p* **
	**Median/*n***	**(IQR/%)**	**Median/*n***	**(IQR/%)**		**Median/*n***	**(IQR/%)**	**Median/*n***	**(IQR/%)**	
Age	44.0	(34.5–52.5)	42.5	(35.3–45.3)	0.542	43.0	(36.0–52.0)	37.0	(31.8–47.3)	0.102
LPL (ng/mL)	65.8	(51.2–77.2)	62.5	(42.8–86.6)	0.506	69.7	(50.3–87.3)	46.1	(39.0–57.6)	** 0.039 * **
GPIHBP1 (pg/mL)	989.7	(728.1–1095.6)	950.7	(755.4–1093.7)	0.667	968.7	(777.9–1098.2)	827.1	(693.5–1107.3)	0.641
HTGL (ng/mL)	38.0	(31.2–54.6)	45.8	(24.8–65.1)	0.969	38.0	(27.1–57.9)	40.6	(36.7–89.8)	0.380
IRI (µU/mL)	3.9	(2.5–5.8)	11.8	(8.6–25.6)	** <0.001 * **	5.7	(3.7–8.8)	7.5	(5.3–26.6)	0.232
FRTN (ng/mL)	90.5	(55.1–108.4)	92.3	(63.5–373.2)	0.213	90.8	(63.1–146.9)	76.7	(46.1–231.2)	0.546
FPG (mg/dL)	93.0	(89.0–98.5)	95.5	(89.3–116.3)	0.449	95.0	(89.0–99.0)	99.0	(91.8–106.3)	0.174
HOMA-IR	0.9	(0.6–1.4)	2.7	(1.9–9.9)	** <0.001 * **	1.4	(0.9–2.0)	1.9	(1.2–7.4)	0.232
HOMA-β	50.1	(35.4–63.2)	133.0	(101.3–167.1)	** <0.001 * **	62.2	(39.3–105.2)	73.0	(65.9–210.1)	0.192
TG (mg/dL)	82.0	(57.0–126.5)	135.5	(104.5–240.5)	** 0.008 * **	104.0	(59.0–140.0)	116.5	(83.5–188.3)	0.232
HDL-C (mg/dL)	56.5	(43.0–65.0)	48.0	(41.0–56.0)	0.340	56.0	(47.0–65.0)	32.0	(29.3–36.3)	** <0.001 * **
LDL-C (mg/dL)	122.0	(92.5–139.5)	111.5	(105.3–118.0)	0.345	117.0	(105.0–139.0)	109.5	(92.0–143.3)	0.733
Ht (cm)	173.0	(168.0–177.0)	175.0	(167.5–180.0)	0.699	173.0	(168.0–177.0)	175.0	(171.8–180.0)	0.384
BW (kg)	67.8	(63.4–75.7)	85.5	(59.6–94.1)	0.187	70.0	(60.3–79.3)	75.7	(66.6–85.0)	0.414
BMI (kg/m^2^)	22.8	(20.8–25.4)	27.0	(22.1–29.5)	** 0.040 * **	24.0	(20.7–27.4)	25.1	(22.2–26.8)	0.694
Body fat (%)	20.7	(17.6–24.7)	29.5	(23.2–33.9)	** 0.010 * **	25.0	(17.2–30.5)	22.3	(20.5–25.4)	0.813
WC (cm)	81.8	(78.9–88.4)	95.3	(82.4–107.1)	** 0.020 * **	83.2	(79.0–92.8)	87.3	(81.2–97.1)	0.655
Diabetes	0	(0.0)	1	(100.0)	0.179	1	(4.3)	0	(0.0)	0.603
Hypertension	2	(9.5)	4	(33.3)	0.088	4	817.4)	1	(16.7)	0.967
TG ≥ 150 (mg/dL)	2	(9.5)	6	(50.0)	** 0.009 * **	5	(21.7)	2	(33.3)	0.554
BMI ≥ 25 (kg/m^2^)	6	(28.6)	8	(66.7)	** 0.033 * **	10	(43.5)	3	(50.0)	0.775

Statistically significant differences were indicated with a *p* < 0.05 value and marked with a bold red and *.

**Table 5 nutrients-17-03207-t005:** Multivariate and univariate analyses with HDL-C < 40 as the objective variable.

		Multivariate Analysis			Univariate Analysis		
		aOR	(95% CI)	*p*	cOR	(95% CI)	*p*
Control	LPL (ng/mL)	0.90	(0.76–1.06)	0.188	0.91	(0.80–1.04)	0.155
	HOMA-IR	0.18	(0.00–12.44)	0.424	0.98	(0.61–1.58)	0.930
	HOMA-β	1.06	(0.93–1.20)	0.393	1.00	(0.99–1.01)	0.977
	BMI	1.24	(0.72–2.12)	0.436	1.25	(0.86–1.83)	0.243
PLH	LPL (ng/mL)	0.90	(0.81–1.00)	** 0.044 * **	0.92	(0.84–1.00)	0.055
	HOMA-IR	0.63	(0.24–1.70)	0.366	1.12	(0.92–1.37)	0.246
	HOMA-β	1.01	(0.97–1.06)	0.618	1.01	(1.00–1.02)	0.157
	BMI	1.03	(0.67–1.60)	0.882	1.00	(0.81–1.24)	0.975

Both univariate and multivariate analyses were performed by using logistic regression analysis. Those having a probability of significance (*p* < 0.05) are marked with bold red and *.

## Data Availability

The original contributions presented in this study are included in the article/Appendix A. Further inquiries can be directed to the corresponding author.

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
