# Peer review of "Clinical Significance of Lipoprotein Lipase (LPL) in People Living with HIV: A Comprehensive Assessment Including Lipidemia, Body Composition, Insulin Secretion, and Insulin Resistance"

_nutrients, 2025, doi:10.3390/nu17203207_

Round 1
Reviewer 1 Report
Comments and Suggestions for Authors
The authors describe their work on the clinical significance of lipoprotein lipase (LPL) in people living with HIV (PLH). It was found that compared to non-HIV controls, PLH have insulin resistance characterized by low HDL-C and LPL. It was concluded that measurement of LPL may confer the risk assessment and decision-making with relevance to antilipemic drugs in PLH. Additionally, insulin resistance may affect HIV antiviral therapy. This is an interesting study. Appropriate methodology has been employed and the conclusions appear to be justified based on the data at hand. The authors have described some limitations of their work, however; I have a few recommendations for consideration.
- Introduction. Can the authors provide a stronger rationale for why the study was undertaken as well as provide a clear hypothesis to be tested in the study?
- Methods/Study design. The study would have been stronger with inclusion of both premenopausal and postmenopausal women.
- Methods/Results. Was/can LPL activity be measured in the PLH patient population?
- Discussion. The authors should elaborate and emphasize the novelty aspect of their work and expand on the clinical applicability of their findings.
Author Response
Answer to Reviewer 1
Comments and Suggestions for Authors
The authors describe their work on the clinical significance of lipoprotein lipase (LPL) in people living with HIV (PLH). It was found that compared to non-HIV controls, PLH have insulin resistance characterized by low HDL-C and LPL. It was concluded that measurement of LPL may confer the risk assessment and decision-making with relevance to antilipemic drugs in PLH. Additionally, insulin resistance may affect HIV antiviral therapy. This is an interesting study. Appropriate methodology has been employed and the conclusions appear to be justified based on the data at hand. The authors have described some limitations of their work, however; I have a few recommendations for consideration.
- Introduction. Can the authors provide a stronger rationale for why the study was undertaken as well as provide a clear hypothesis to be tested in the study?
Answer: Thank you for your valuable comments. We have reported that insulin resistance leads to a decrease in LPL and an increase in GPIHBP1 levels [14]. The level of LPL in peripheral bloodstream tends to be lower in individuals with more risk factors for metabolic syndrome, such as impaired glucose tolerance, dyslipidemia, and hypertension. This study was conducted to determine whether, in the PLH, LPL levels are also low, as seen in metabolic syndrome. We added the following text to lines 71-76 and 82-83 of the introduction. Line 71-76: “We have reported that insulin resistance leads to a decrease in LPL and an increase in GPIHBP1 levels [14]. The level of LPL in peripheral bloodstream tends to be lower in individuals with more risk factors for metabolic syndrome, such as impaired glucose tolerance, dyslipidemia, and hypertension. In the case of PLH, it is unclear whether low LPL levels are also observed, as seen in metabolic syndrome.” Line 82-83: “In addition, we analyzed the relationship between the effectiveness of HIV antiviral therapy and glucose and lipid metabolism.”
- Methods/Study design. The study would have been stronger with inclusion of both premenopausal and postmenopausal women.
Answer: Thank you for your valuable comments. We have reported that the relationship between blood LPL levels and insulin resistance differs between males and females [14]. The research budget is limited, making it impossible to analyze data from female participants. Please understand our situation. We would like to secure research funding and conduct further analysis of women's PLH in the future. We added the following text to lines 476-478 of the limitation: “Fifth, no analysis of data from PLH female was conducted. There are gender differences in LPL levels in peripheral bloodstream and insulin resistance. Future analysis of females with PLH is needed.”
- Methods/Results. Was/can LPL activity be measured in the PLH patient population?
Answer: Thank you for your valuable comments. Intravenous administration of heparin is necessary to measure LPL activity. Intravenous heparin administering is an invasive procedure that exposes research participants to the risk of bleeding. We have reported that fasting blood levels of LPL reflect LPL activity after heparin administration [Shirakawa et al. Clin Chim Acta. 2015, 440: 193-200.]. This study ensures the safety of the research participants by not administering heparin and by analyzing LPL levels without any invasive procedures. This fact represents an important step that aligns with the Helsinki Declaration. Please understand this situation.
Reference
Shirakawa T, Nakajima K, Shimomura Y, Kobayashi J, Stanhope K, Havel P, Machida T, Sumino H, Murakami M. Comparison of the effect of post-heparin and pre-heparin lipoprotein lipase and hepatic triglyceride lipase on remnant lipoprotein metabolism. Clin Chim Acta. 2015 Feb 2:440:193-200. doi: 10.1016/j.cca.2014.07.020.
- Discussion. The authors should elaborate and emphasize the novelty aspect of their work and expand on the clinical applicability of their findings.
Answer: Thank you for your valuable comments. With the development of antiretroviral therapies for HIV, the average lifespan of PLH has increased, but this has led to a growing problem of atherosclerotic diseases among this population. This study revealed that in PLH, HDL-C and LPL levels were lower compared to the control group. Low levels of HDL-C and LPL reflect increased insulin resistance. In this study, no significant difference in HOMA-IR, one of the indicators of insulin resistance, was observed between the control group and the PLH group. While the HOMA-β value, which reflects basal insulin secretion, was significantly higher in the PLH group. As shown in Table 2a, PLH with high HOMA-β values also had significantly higher HOMA-IR, fasting insulin, TG and WC levels and significantly lower LPL levels. These findings suggest that in PLH, those with high HOMA-β values exhibit increased basal insulin secretion as a compensatory mechanism to counteract insulin resistance. Supporting this, it has been reported that HIV antiviral therapy increases insulin resistance [28, 29]. In addition, it has been reported that LPL mRNA expression levels decrease [29]. Taken together, this study showed that even when classical lipid markers were within clinically normal ranges, the LPL measurement may confer risk assessment and decision-making with relevance to ALD in PLH. Additionally, low levels of HIV-RNA were detected in the high HOMA-β group. These indicate that HIV antiviral therapy and glucose metabolism may interact with each other. We made the following changes to the text.
- We added the following text to lines 218-221 of the Results: “These findings suggest that in PLH, those with high HOMA-β values exhibit increased basal insulin secretion as a compensatory mechanism to counteract insulin resistance. Increased insulin is needed to maintain proper blood glucose levels.”
- 2. We have amended lines 357-361 of the discussion as follows: “Therefore, even when classical lipid markers were within clinically normal ranges, the LPL measurement may confer risk assessment and decision-making with relevance to ALD in PLH. Additionally, low levels of HIV-RNA were detected in the high HOMA-β group. These indicate that HIV antiviral therapy and glucose metabolism may interact with each other.”
- 3. We have amended lines 482-484 of the conclusion as follows: “Low HDL-C and low LPL levels in PLH are likely due to two mechanisms: a direct effect of ARVs on reducing LPL expression, and an indirect effect mediated by increased insulin resistance.”
Reviewer 2 Report
Comments and Suggestions for Authors
The authors conducted a cross-sectional, single-center, non-interventional study evaluating clinical significance of lipoprotein lipase, HDL-c and insulin resistance in people living with HIV. The topic is of great interest, however, the relatively small sample size and somewhat overstated interpretation weaken the strength of the conclusions. There are several issues to be further addressed to strengthen this study.
- The study included only 48 PLH (33 as age&sex matched), which seems relatively small to support multiple regression analyses. Please justify the statistical power and stability.
- Multiple logistic regression models and comparisons were performed, but there is no indication that corrections for multiple testing (e.g., Bonferroni, FDR) were applied. This raises the possibility of false-positive findings.
- In the abstract, only HOMA-β is mentioned but meanwhile insulin resistance is indicated. However, HOMA-β primarily reflects β-cell secretory function, whereas HOMA-IR is the conventional measure of insulin resistance. The interpretation appears somewhat confusing and should be clarified.
- The observation that low HIV-RNA levels were detected in the high HOMA-β group is intriguing, but no potential mechanism or clinical relevance is discussed. Could this reflect ART regimen effects or immune status differences?
- Although LPL predicted HDL-C <40 mg/dL with high sensitivity, the specificity was relatively modest (60.9%). Since HDL-c itself being considered biomarker for atherosclerosis and CVD, how clinically useful is LPL as a predictive biomarker in practice, given the trade-off between sensitivity and specificity?
- The statement “insulin resistance characterized by low HDL-C and LPL” may be misleading, since low HDL-C and low LPL are more metabolic abnormalities than direct markers of insulin resistance.
- The conclusion “Insulin resistance may affect HIV antiviral therapy” seems overstated given the limited data. It would be advisable to tone down this claim.
- In Figure 2f, p=0.047 when comparing the 1st column to the 3rd, yet p=0.126 when comparing the 2nd to the 4th, it is strange because one may expect an even lower p value for the later judging by the figure. Is the result/illustration right? Please confirm.
- It is my own opinion but could you use “*” in the table to indicate significance? The sign “†”, when it is small, looks like “↑”(indicates increase), causes confusion.
- Table 2 is relatively large. Could you extract the key finding and present it as a figure, besides the table itself?
- Figure 3, ROC, when you put your mouse onto the figure, some Japanese characters will pop up, saying the figures generated by AI have the chance to be wrong. I think it is acceptable to feed AI with data and get the ROC it generated, but you should pay attention and state it in the Materials and Methods section.
- The labels in most figures are too small, especially the y-axis labels. Please enlarge them to make it easy for the readers.
- Similarity and differences of the current study to previous published studies should be more extensively discussed.
Author Response
Answer to Reviewer 2
Comments and Suggestions for Authors
The authors conducted a cross-sectional, single-center, non-interventional study evaluating clinical significance of lipoprotein lipase, HDL-c and insulin resistance in people living with HIV. The topic is of great interest, however, the relatively small sample size and somewhat overstated interpretation weaken the strength of the conclusions. There are several issues to be further addressed to strengthen this study.
- The study included only 48 PLH (33 as age & sex matched), which seems relatively small to support multiple regression analyses. Please justify the statistical power and stability.
Answer: Thank you for your valuable comments. As you mention, this study was performed in a single-center, small population and cross-sectional settings with limited number of participants. We modified the following text to lines 463-467 of the limitation: “First, this is a single-center observational study, so that findings from our statistical analysis have a risk of false positive due to the limited number of participants. No multiplicity adjustment was applied in the main analysis. Further research with larger number of participants in multi-center is needed to elucidate causal relationships.”
- Multiple logistic regression models and comparisons were performed, but there is no indication that corrections for multiple testing (e.g., Bonferroni, FDR) were applied. This raises the possibility of false-positive findings.
Answer: Thank you for your valuable comments. The primary hypothesis of this study was based on a limited number of pre-specified clinical variables. Therefore, no multiplicity adjustment was applied in the main analysis. As you mention, the p-values calculated in the ROC analyses (Figures 3. and 4.) conducted based on the multivariate and univariate analysis results in Table 3. may not be independently significant. They should be interpreted solely as meaningful reference values within the context of multifactorial associations, with reference to the AUC and accompanying 95% confidence intervals. This point has been added as a limitation. We modified the following text to lines 463-467 of the limitation: “First, this is a single-center observational study, so that findings from our statistical analysis have a risk of false positive due to the limited number of participants. No multiplicity adjustment was applied in the main analysis. Further research with larger number of participants in multi-center is needed to elucidate causal relationships.”
- In the abstract, only HOMA-β is mentioned but meanwhile insulin resistance is indicated. However, HOMA-β primarily reflects β-cell secretory function, whereas HOMA-IR is the conventional measure of insulin resistance. The interpretation appears somewhat confusing and should be clarified.
Answer: Thank you for your valuable comments. This study revealed that in PLH, HDL-C and LPL levels were lower compared to the control group. Low levels of HDL-C and LPL reflect increased insulin resistance. In this study, no significant difference in HOMA-IR, one of the indicators of insulin resistance, was observed between the control group and the PLH group. On the other hand, the HOMA-β value, which reflects basal insulin secretion, was significantly higher in the PLH group. As shown in Table 2a, PLH with high HOMA-β values also had significantly higher HOMA-IR, fasting insulin, TG, and WC levels and significantly lower LPL levels. These findings suggest that in PLH, those with high HOMA-β values exhibit increased basal insulin secretion as a compensatory mechanism to counteract insulin resistance. Increased insulin is needed to maintain proper blood glucose levels. These facts suggest that the high HOMA-β in PLH in this study is due to increased insulin resistance We modified the discussion, conclusion and abstract.
Abstract. Line 41-45: “In Japanese individuals, compared to non-HIV controls, PLH has low HDL-C and LPL. The measurement of LPL may confer the risk assessment and decision-making with relevance to ALD in PLH. Additionally, the effectiveness of HIV antiviral therapy and glucose tolerance may interact with each other.”
Results. Line 218-221: “These findings suggest that in PLH, those with high HOMA-β values exhibit increased basal insulin secretion as a compensatory mechanism to counteract insulin resistance. In-creased insulin is needed to maintain proper blood glucose levels.”
Discussion. Line 372-378: “In PLH, patients with high HOMA-β levels were characterized by high insulin resistance, as indicated by high HOMA-IR, fasting insulin, body fat mass, WC, BMI, and TG, and low LPL and HDL-C. These findings suggest that in PLH, those with high HOMA-β values exhibit increased basal insulin secretion as a compensatory mechanism to counteract insulin resistance. Increased insulin is needed to maintain proper blood glucose levels. These facts suggest that the high HOMA-β in PLH in this study is due to increased insulin resistance.”
Conclusion. Line 480-481: “In Japanese individuals, compared to non-HIV controls, PLH had lower LPL and HDL-C, and higher HOMA-β.”
- The observation that low HIV-RNA levels were detected in the high HOMA-β group is intriguing, but no potential mechanism or clinical relevance is discussed. Could this reflect ART regimen effects or immune status differences?
Answer: Thank you for your valuable comments. Based on the above facts (answer to the comment 3), we believe that the high HOMA-β in PLH in this study is due to increased insulin resistance. Therefore, it is thought that increased insulin resistance affected immune function, resulting in low levels of HIV viral load. Alternatively, it is possible that ART treatment is not effective enough, leaving low levels of HIV virus remaining, resulting in increased insulin resistance. We added a description about this idea to line 386-389 in Discussion: “Furthermore, in human adipose tissue, the integrase inhibitors dolutegravir (DTG) and raltegravir (RAL) induce insulin resistance [28] and suppress LPL mRNA expression [29]. Both DTG and RAL increased the interleukin-6 (IL-6) gene expression [28, 29]. Taken together with our findings, the presence of detectable HIV-RNA under ART may be associated with increased accumulation of visceral fat, insulin resistance, and IL-6 and decreased mRNA expression of LPL in adipose tissue.”
- Although LPL predicted HDL-C <40 mg/dL with high sensitivity, the specificity was relatively modest (60.9%). Since HDL-C itself being considered biomarker for atherosclerosis and CVD, how clinically useful is LPL as a predictive biomarker in practice, given the trade-off between sensitivity and specificity?
Answer: Thank you for your valuable comments. Currently, no therapeutic drugs that increase HDL-C are in development or available. Most of the HDL-C levels of PLH shown in Table 1 are 40 mg/dL or higher and are not classified as having low HDL-C. Therefore, although the HDL-C level in the PLH group was significantly lower than that in the healthy group, it was not at a level that required consideration of medication. This fact suggests that in the PLH group, even if HDL-C is within the reference range, if LPL levels are low, in addition to improving lifestyle habits such as diet and exercise, drug treatment may be considered. We made some revisions to clarify this matter.
Abstract. Line 41-45: “In Japanese individuals, compared to non-HIV controls, PLH has low HDL-C and LPL. The measurement of LPL may confer the risk assessment and decision-making with relevance to ALD in PLH. Additionally, the effectiveness of HIV antiviral therapy and glucose tolerance may interact with each other.”
Discussion. Line 357-359: “Therefore, even when classical lipid markers were within clinically normal ranges, the LPL measurement may confer risk assessment and decision-making with relevance to ALD in PLH.”
Discussion. Line 410-412: “Even when classical lipid markers were within clinically normal ranges, the LPL measurement contributes risk assessment and decision-making with relevance to ALD in PLH.”
- The statement “insulin resistance characterized by low HDL-C and LPL” may be misleading, since low HDL-C and low LPL are more metabolic abnormalities than direct markers of insulin resistance.
Answer: Thank you for your valuable comments. As you mentioned, low HDL-C and low LPL are both independent factors associated with insulin resistance. So that we repaired the description in abstract and discussion.
Abstract. Line 41-43: “In Japanese individuals, compared to non-HIV controls, PLH has low HDL-C and LPL.
Conclusion. Line 480-481: “In Japanese individuals, compared to non-HIV controls, PLH had lower LPL and HDL-C, and higher HOMA-β.”
- The conclusion “Insulin resistance may affect HIV antiviral therapy” seems overstated given the limited data. It would be advisable to tone down this claim.
Answer: Thank you for your valuable comments. We avoided asserting a causal relationship between antiretroviral therapy and insulin resistance, instead modifying the description to emphasize the interplay between treatment and glucose metabolism. We modified the discussion and abstract.
Abstract. Line 44-45: “Additionally, the effectiveness of HIV antiviral therapy and glucose tolerance may interact with each other.”
Discussion. Line 359-361: “Additionally, low levels of HIV-RNA were detected in the high HOMA-β group. These indicate that HIV antiviral therapy and glucose metabolism may interact with each other.”
Discussion. Line 386-389: “Furthermore, in human adipose tissue, the integrase inhibitors dolutegravir (DTG) and raltegravir (RAL) induce insulin resistance [28] and suppress LPL mRNA expression [29]. Both DTG and RAL increased the interleukin-6 (IL-6) gene expression [28, 29]. Taken together with our findings, the presence of detectable HIV-RNA under ART may be associated with increased accumulation of visceral fat, insulin resistance, and IL-6 and decreased mRNA expression of LPL in adipose tissue.”
- In Figure 2f, p=0.047 when comparing the 1st column to the 3rd, yet p=0.126 when comparing the 2nd to the 4th, it is strange because one may expect an even lower p value for the later judging by the figure. Is the result/illustration right? Please confirm.
Answer: Thank you for your valuable comments. We recalculated the statistics, and the results in Figure 2 f (p=0.047 when comparing the 1st column to the 3rd, yet p=0.126 when comparing the 2nd to the 4th) were confirmed to be correct. We have attached the raw output of results calculated by SPSS ver30.0 (file name: OUTPUT.docx). Please confirm the attached file.
- It is my own opinion but could you use “*” in the table to indicate significance? The sign “†”, when it is small, looks like “↑”(indicates increase), causes confusion.
Answer: Thank you for your valuable comments. According to the International Committee of Medical Journal Editors' guidelines, medical journals are required to use the order *,†,‡,§,||,¶,**,††,‡‡ for footnotes. Therefore, we have revised the manuscript to use * first for footnotes in this paper. We revised Tables and legends for table.
- Table 2 is relatively large. Could you extract the key finding and present it as a figure, besides the table itself?
Answer: Thank you for your valuable comments. We revised Table 2, extracting only the key points, and the original version of Table 2 was moved to the supplementary data section. Please confirm Table 2a, 2b, Supplemental Table 2a, and 2b.
- Figure 3, ROC, when you put your mouse onto the figure, some Japanese characters will pop up, saying the figures generated by AI have the chance to be wrong. I think it is acceptable to feed AI with data and get the ROC it generated, but you should pay attention and state it in the Materials and Methods section.
Answer: Thank you for comments. We would like to clarify that no artificial intelligence (AI) tools were used in the creation of Figure 3 or any other figures. Furthermore, we have to apologize for stating in the initial manuscript that statistical software” All statistical analyses were performed using Stata 30.0 (StataCorp, College Station, TX, USA) was used”. The actual statistical analyses and graphing were performed using IBM SPSS Statistics version 30.0 without the use of AI-based tools. The Material and Methods section has been revised.
Line 159-161: “All statistical analyses and creating illustrations were performed using IBM SPSS version 30.0.”
- The labels in most figures are too small, especially the y-axis labels. Please enlarge them to make it easy for the readers.
Answer: Thank you for your valuable comments. We repaired the size of labels especially in Figure.2, 3 and 4.
- Similarity and differences of the current study to previous published studies should be more extensively discussed.
Answer: Thank you for your valuable comments. Compared to previous studies, reports from the era when protease inhibitors were widely used for HIV treatment support reduced LPL levels associated with high TG, consistent with our data. Our findings have brought renewed attention to the clinical significance of low LPL levels in an era where HIV is mainly treated with integrase inhibitors, and is novelty and difference of previous reports. Furthermore, PLH did not show higher HOMA-IR but higher HOMA-βcompared to controls in our study. However, recent studies in USA showed a higher prevalence of diabetes mellitus in PLH [24], and studies in China showed a value of triglyceride-glucose (TyG) index, an emerging marker of insulin resistance to predict the incidence of type 2 diabetes mellitus (T2DM) [25]. These conflicts between them and our results suggest the difference in genetic background due to ethnicity and lifestyle especially in food. Therefore, we emphasized that our data represents “Japanese evidence” and the necessity of further study in “across the country”.
We have added the following text to the discussion (line 378-382): “Recent studies in USA showed a higher prevalence of diabetes mellitus in PLH [24], and studies in China showed a value of triglyceride-glucose (TyG) index, an emerging marker of insulin resistance to predict the incidence of type 2 diabetes mellitus (T2DM) [25]. These conflicts between them and our results suggest the difference in genetic back-ground due to ethnicity and lifestyle especially in food.”
We have revised the Conclusion as follows (line 480-489): “In Japanese individuals, compared to non-HIV controls, PLH had lower LPL and HDL-C, and higher HOMA-β. Our findings highlight the potential role of LPL as a key marker in lipid metabolism among PLH, providing novel insights into their long-term management. Low HDL-C and low LPL levels in PLH are likely due to two mechanisms: a direct effect of ARVs on reducing LPL expression, and an indirect effect mediated by increased insulin resistance. Furthermore, the results of this study suggested the association of insulin resistance and the control of HIV-RNA under antiviral therapy. Further studies are needed across the country to examine the efficacy of therapeutic interventions targeting the LPL activity to improve the risks of cardiovascular events in PLH.”
Round 2
Reviewer 1 Report
Comments and Suggestions for Authors
The authors have addressed all initial concerns and have adequately revised their manuscript. I have no further comments.
Author Response
Thank you very much.
Reviewer 2 Report
Comments and Suggestions for Authors
This study has been improved after revision. Although the main concerns of relatively small sample size can not be solved, the study is of interest and is in general well presented. It could be considered for publication.
Author Response
Thank you very much.